# Morphological and Molecular Analyses of the Interaction between *Rosa multiflora* and *Podosphaera pannosa*

**DOI:** 10.3390/genes13061003

**Published:** 2022-06-02

**Authors:** Ying Bao, Xue Zhang, Xiaoxiang Sun, Manzhu Bao, Yuanyuan Wang

**Affiliations:** 1Key Laboratory of Horticultural Plant Biology of Ministry of Education, College of Horticulture and Forestry Sciences, Huazhong Agricultural University, Wuhan 430070, China; baoying090924@126.com (Y.B.); zhangxue19881228@163.com (X.Z.); sunxx328@163.com (X.S.); mzbao@mail.hzau.edu.cn (M.B.); 2Department of Life Science, Tangshan Normal University, Tangshan 063000, China

**Keywords:** *Rosa multiflora*, powdery mildew, cytological observation, disease-resistance related gene

## Abstract

Powdery mildew disease caused by *Podosphaera*
*pannosa* is the most widespread disease in global cut-rose production, as well as a major disease in garden and pot roses. In this study, the powdery mildew resistance of different wild rose varieties was evaluated. Rose varieties with high resistance and high sensitivity were used for cytological observation and transcriptome and expression profile analyses to study changes at the morphological and molecular levels during the interaction between *Rosa multiflora* and *P. pannosa*. There were significant differences in powdery mildew resistance among three *R. multiflora* plants; *R. multiflora* ‘13’ had high resistance, while *R. multiflora* ‘4’ and ‘1’ had high susceptibility. Cytological observations showed that in susceptible plants, 96 and 144 h after inoculation, hyphae were observed in infected leaves; hyphae infected the leaf tissue through the stoma of the lower epidermis, while papillae were formed on the upper epidermis of susceptible leaf tissue. Gene ontology enrichment analysis showed that the differentially expressed genes that were significantly enriched in biological process functions were related to the secondary metabolic process, the most significantly enriched cellular component function was cell wall, and the most significantly enriched molecular function was chitin binding. Changes in the transcript levels of important defense-related genes were analyzed. The results showed that *c**hitinase* may have played an important role in the interactions between resistant *R. multiflora* and *P. pannosa*. Jasmonic acid and ethylene (JA/ET) signaling pathways might be triggered in the interaction between susceptible *R. multiflora* and *P. pannosa*. In the resistant *R. multiflora*, the salicylic acid (SA) signaling pathway was induced earlier. Between susceptible plants and resistant plants, key phenylpropanoid pathway genes were induced and upregulated after *P. pannosa* inoculation, demonstrating that the phenylpropanoid pathway and secondary metabolites may play important and active roles in *R. multiflora* defense against powdery mildew infection.

## 1. Introduction

Roses (*Rosa* sp.) have become increasingly important to the floriculture industry as ornamental plants [1]. Rose powdery mildew, caused by *Podosphaera pannosa*, is one of the most serious and widespread rose diseases [2,3,4]. Rose powdery mildew affects almost all plant tissues and can lead to leaf distortion, curling, premature defoliation, and abnormal flowers [5]. These symptoms reduce cut-flower production in both the greenhouse and the field and result in significant economic losses [6]. At present, powdery mildew is typically managed through the use of synthetic fungicides [5], which increase production costs, cause environmental pollution, and may lead to pathogen resistance [7]. Eco-friendly production methods for plant disease suppression need to be developed. Therefore, it is of great significance to explore powdery mildew resistance resources and resistance-related genes from the existing rose germplasm resources and to cultivate powdery mildew-resistant varieties [8].

Plant cells are generally protected against pathogen infection by several types of chemical and structural barriers, including phytoalexins, cell walls, and phenolic compounds [9,10]. The synthesis of phenols is a particularly important component of plant defense mechanisms against pathogenic microbes [11], and many of these phenols are derived from the phenylpropanoid metabolic grid. Recent studies have demonstrated that secondary metabolites in the phenylpropanoid pathway, such as phytoalexins, phenolic acids, flavonoids, and lignin, are important for plant defenses against pathogenic infections [12,13]. Plants also enhance cell walls by depositing cell wall-strengthening materials, such as callose and lignin [14].

Plants respond and adapt to biotic and abiotic stress at the cellular, molecular, and physiological levels [15]. Plant disease resistance depends on the expression of many defense genes and the coordination of products so that the plant effectively resists the infection of pathogenic bacteria; thus, pathogens not only trigger local defense responses, but also induce signal production [16]. In many plant species, salicylic acid (SA) plays a crucial role as an endogenous regulator of localized and systemic acquired resistance (SAR) to pathogen infection [17]. SAR is an induced defense mechanism that confers long-lasting protection against a broad spectrum of pathogenic microorganisms [18]. Biochemical and transcriptional studies of SAR have revealed that this process involves the transient accumulation of SA and the induction of defense-related genes [19]. In addition, previous studies showed that the accumulation of SA was accompanied by the increase of phenolic compounds, as well as callose and lignin deposition, which indicated that SA was related to the modification of the structural cell wall [20].

In addition, many studies have found that induced disease resistance has three main signaling pathways in plants: the SA-regulated defense response, the jasmonic acid (JA)-regulated defense response, and the ethylene (ET)-regulated defense response [21]. Plants may therefore use different signaling pathways to initiate the most effective defense against a particular pathogen [22,23,24].

However, relatively few studies have compared the morphological or transcription-level responses to rose powdery mildew between mildew-resistant and mildew-susceptible rose varieties. In this study, the responses to powdery mildew were compared between two different wild rose varieties. This study focused on characterizing changes at the morphological and molecular levels during the interaction between the wild rose varieties and powdery mildew. In order to examine the RNA levels of plant defense-related genes after powdery mildew infection, high-throughput sequencing was used to screen the *R**osa multiflora* transcriptome in the process of infection with powdery mildew, and the genes with powdery mildew resistance were identified. The temporal and spatial expression levels of these genes were analyzed using real-time polymerase chain reaction (PCR). These results provide a framework for the study of the mechanisms underlying the resistance to powdery mildew in roses.

## 2. Materials and Methods

### 2.1. Morphological Observations and Molecular Identification of Powdery Mildew

Three *R**osa*
*multiflora* plants, denoted ‘1’, ‘4’, and ‘13’, were collected from the Shennongjia Nature Reserve in central China’s Hubei province. They were planted in the rose germplasm resource nursery of Huazhong Agricultural University, Wuhan, China, for preservation. The leaves infected with powdery mildew of *Rosa chinensis* and *R. multiflora* were collected and placed into centrifugal tubes filled with fixed liquid (glacial acetic acid (V):absolute ethanol (V) = 1:3). After 22 h of fixation, the fixation solution was poured out. The centrifugal nozzle was sealed with gauze and rinsed under running water for 4 h to fully remove the residual fixed liquid. Then, a pipette gun was used to remove the residual liquid from the centrifugal tube, and the preservation liquid (lactic acid (V):water (V):glycerol = 1:1:1) was injected into the centrifugal tube, then stored at 4 °C. Then, a temporary slide was set up for observation and stained with staining solution (0.2 g aniline blue was added to 100 mL distilled water and diluted with twice the volume of absolute ethanol before use) during observation.

The DNA of powdery mildew was extracted using the CTAB method. The primers used in this experiment were ITS1 (5′-TCCGTAGGTGAACCTGCGG-3′) and ITS4 (5′-TCCTCCGCTTATTGATATGC-3′), which were used as universal primers for the ITS region of fungal ribosomes [25]. The PCR reaction system consisted of 5 μL 10 × KOD buffer, 3 μL MgSO_4_, 5 μL DNTPs, 1 μL cDNA, 1.5 μL each of upstream and downstream primers, 1 μL KOD plus high-fidelity enzyme, and sterile ddH_2_O to a total volume of 20 μL. The PCR reaction procedure was as follows: 94 °C for 5 min, 94 °C for 30 s, 56.0 °C for 30 s, and 68 °C for 1 min, repeated for 35 cycles, with a final step of 68 °C for 5 min. The PCR products were detected by agarose gel electrophoresis, and the target bands were recovered using the TaKaRa Mini BESTAgarose Gel DNA Extraction Kit (TaKaRa, Dalian, China) and connected with pMD18-T vector. The products were transformed into Escherichia coli, the blue and white spots were screened, and the white single colonies were selected for identification. According to the PCR results, the identified positive clones were sent to Sangon Biotech (Shanghai) Co., Ltd., Shanghai, China, for sequencing. The sequences were compared on the NCBI website (http://www.ncbi.nlm.nih.gov/, accessed on 10 June 2018).

### 2.2. Assessment of R. multiflora Resistance to Powdery Mildew

*R. multiflora* collected from the Shennongjia Nature Reserve in central China’s Hubei province were employed in this research. From September 2018 to June 2019, the disease progression of three *R. multiflora* plants, denoted ‘1’, ‘4’, and ‘13’, was observed and recorded every 15 d in the field according to the grading standard shown in Table 1. The disease index and resistance index of the population were calculated according to 20 observed results.

The resistance of *R. multiflora* to powdery mildew was also determined based on microscopic observations. The grade standards used in the microscopic observation method are shown in Table 2. Spores from *P. pannosa* that infected roses were employed as an inoculum. *P. pannosa* spores in a solution of 0.1% Tween 20 (the spore concentration was adjusted to 10^−^^4^–10^−^^5^·mL^−1^) were applied to whole plants using a sprayer. Five leaves were collected from each *R. multiflora* variety at 24 h and 96 h post-inoculation (hpi), then placed into centrifuge tubes containing stationary liquid. Temporary slides were set up for observation according to the above methods. The observations included the mycelial growth rate (24 hpi) and the mycelial coverage area and conidiophore formation grade (96 hpi). The experiment was repeated three times. The disease index = Σ(number of diseased leaves × disease grade)/(total number of leaves) × most severe disease (grade), and resistance index = 1 − disease index. The average value of these two indexes was used as the final basis for evaluating the disease resistance of *R. multiflora*. The degree of disease resistance could be divided into five categories: immune (I), high resistance (HR), moderate resistance (MR), moderate sensitivity (MS), and high sensitivity (HS), with resistance indexes of 1.00, 0.80–1.00, 0.40–0.80, 0.20–0.40, and less than 0.20, respectively.

In order to further evaluate the resistance of *R**. multiflora* to powdery mildew, the cutting seedlings of *R. multiflora* ‘4’ and ‘13’ were inoculated with powdery mildew. Hard branches with good growth and without diseases and insect pests were selected to obtain cuttings, and one rooted cutting was grown in each pot. Young plants with three to five leaves (leaf age 20–35 d) that exhibited good growth without diseases and pests were used in the following experiments. No fungicide was used during the growth stage of the tested leaves. *P. pannosa* spores were employed as inoculum. The powdery mildew inoculation method is shown above 2.2. Each treatment included seven rooted cuttings of each line, with three replicates. Ten days after inoculation, according to the percentage of the total leaf area occupied by pathogens, grades 0–6 were used to describe the development of the disease. The evaluation criteria are shown in Table 3.

### 2.3. Cytological Studies

According to the research results shown in Section 2.2, *R. multiflora* ‘4’ was highly susceptible, while *R. multiflora* ‘13’ was highly resistant to powdery mildew. The stem segments with nodes of these two plants were washed under tap water for 30 min, surface-sterilized in 70% ethanol for 30 sec and 0.1% HgCl_2_ for 15 min, rinsed three times in sterile distilled water, dried, and then used as explants. The optimum medium for the in vitro growth and multiplication of *R. multiflora* from nodal segments was MS + 6-BA1.0 mg/L + NAA 0.01 mg/L + 30 g/L sucrose + 7.5 g/L agar. The explants were placed in an incubator at 23 °C (±2 °C), relative humidity 60%, with 1500–2000 lux light intensity until sampled.

Pathogen spores from *P. pannosa* were employed as an inoculum. The powdery mildew inoculation method is shown above in (Section 2.2 Assessment of *R. multiflora* resistance to powdery mildew). Sterile water was applied in the non-inoculated treatments. For microscopic studies, the leaves of tissue-culture seedlings were sampled at 0, 12, 24, 48, 96, and 144 hpi. The leaves were gently washed and cut into small pieces (0.2 cm × 0.2 cm). The leaf pieces were fixed in FAA (formalin:acetic acid:70% ethyl alcohol = 1:1:18) standard fixative. The slices were obtained with a Leica RM2265 slicer (Leica, Germany), spread on the slides with the addition of distilled water, and then stained with 1% toluidine blue stain. The stain was rinsed off for about 3 min before baking the slices, and then semi-thin slices were made for the cytological observation of the host–powdery mildew interaction. The methods used followed those described by Li [26].

### 2.4. De Novo Transcriptome Assembly

Resistant and susceptible plants were separately inoculated with a spore suspension of powdery mildew when the seedlings had grown 6–10 leaves. Sterile water was used in the control treatment. Leaf samples were taken from the disease-susceptible and disease-resistant plants at 24 h and 96 h. Total RNA was extracted using an OminiPlant RNA kit (Cwbio, Beijing, China) according to the manufacturer’s instructions, with an additional sonication step after the addition of RLT buffer (Qiagen). Strand-specific RNA libraries were constructed and sequenced on a HiSeq 2500 system (Illumina), according to the manufacturer’s instructions. The samples were analyzed using high-throughput transcription group and express spectrum sequencing by Shanghai Genergy Biotechnology Company (Shanghai, China). Bioinformatics analyses were conducted by Guangzhou Kidio Biotechnology Co., Ltd. (Guangzhou, China).

### 2.5. Gene Annotation and Function Analyses

To annotate the identified genes, the sequences were analyzed with various nucleotide and protein databases, including the NR (NCBI non-redundant protein), NT (NCBI non-redundant transcript), Swiss-Prot, KEGG (Kyoto Encyclopedia of Genes and Genomes), KOG (eukaryotic Ortholog Groups), and GO (Gene Ontology) libraries. The BLASTX algorithm was used with a significance threshold of E-value ≤ 10^−5^. KEGG pathway enrichment analysis of DTGs was performed using KOBAS. GO term enrichment was analyzed using the GOseq R package based on Wallenius non-central hyper-geometric distribution.

### 2.6. Expression of Candidate Defense-Related Genes

Based on the results of gene expression differential analysis and gene function analysis, 22 defense-related genes (*non-race-specific disease resistance* (*NDR1*), *enhanced disease susceptibility* (*EDS1*), *senescence-associated gene 101* (*SAG101*), *ethylene-responsive-element* (*ERF10)*, *kinase**, pathogenesis related protein 1* (*PR1*), *PR5*, *callose synthase* (*CS*), *chitinase*, *catalase* (*CAT*), *peroxidase* (*POD*), *chorismate mutase* (*CM*), *phenylalanine ammonialyase* (*PAL*), *phospholipase D* (*PLD*), *lipoxygenase* (*LOX*), *allene oxide cyclase* (*AOC*), *adenosythomocysteinase* (*Ade*), *ACC* (*1-aminocyclopropane-1-carboxylic*) *synthase* (*ACS*), *ACC oxidase* (*ACO*), *4-coumarate coenzyme A ligase* (*4CL*), *chalcone synthase* (*CHS*), and *cinnamyl alcohol dehydrogenase* (*CAD*)) were screened for gene expression analysis. The expression levels of candidate genes (Appendix A) from the uninoculated and *P. pannosa*-inoculated *R. multiflora* tissue culture seedlings at 12, 24, 48, 96, and 144 hpi were determined with real-time PCR. DNA sequences from candidate *R. multiflora* genes were obtained from the expression profiling analysis. Primer pairs for PCR analyses were designed using Primer5, following the recommendations of Qiagen (Mississauga, Canada). Total RNA was isolated from leaves with the EASY spin rapid RNA extraction kit (Invitrogen, Thermo Fisher Scientific Inc., Waltham, MA, USA), following the manufacturer’s instructions. cDNA was produced by reverse transcription using PrimeScript RT reagent kits with gDNA Eraser (Clontech, Takara Bio, Dalian, China). GADPH primers (Appendix A) were used to standardize RNA samples for real-time PCR.

### 2.7. Statistical Analysis

All experiments were performed using three independent biological replicates, each including three technical replicates. Fold differences in gene expression levels were presented as log2 values and represented the difference between the inoculated and non-inoculated treatments.

## 3. Results

### 3.1. Morphological Observations and Molecular Identification of Powdery Mildew

As shown in Figure 1, the conidia grew in tandem and separated from the conidiophore after maturity, forming an egg shape. The conidiophore was erect, unbranched, and septate. It was preliminarily determined that the powdery mildew found on both Rosa chinensis and *Rosa multiflora* was *P. pannosa*. The amplification results showed that the length of the ITS sequence of the strain was 565 bp. The ITS sequences were compared and analyzed in GenBank, and the homology was 100% with *P. pannosa* (registration number: AB525939.1).

### 3.2. Evaluation of Resistance in R. multiflora

The field incidence of *R. multiflora* ‘1’, ‘4’, and ‘13’ is shown in Figure 2. According to the classification standard, the field disease grade indexes of *R. multiflora* ‘1’, ‘4’, and ‘13’ were recorded, and the disease grade index and resistance index were calculated. The results showed that there were significant differences in powdery mildew resistance among the three plants; *R. multiflora* ‘13’ had high resistance, and *R. multiflora* ‘4’ and ‘1’ had high susceptibility (Table 4).

The leaves of *R. multiflora* ‘1’, ‘4’, and ‘13’ were observed under the microscope. The development process of pathogenic bacteria on the leaves is shown in Figure 3. When inoculated with powdery mildew for 1 d, the conidia germinated, and the average lengths on *R. multiflora* ‘1’, ‘4’, and ‘13’ were 15.2 um, 32.5 um, and 5.6 um, respectively. Four days after inoculation, conidiophores had formed on *R. multiflora* ‘4’; there were some hyphae on *R. multiflora* ‘1’, but no conidiophore was observed; and there were no hyphae on most *R. multiflora* ‘13’, with only some non-germinating conidia observed. Hyphae accounted for 1/20–1/10 and more than 1/10 in the leaves of *R. multiflora* ‘1’ and *R. multiflora* ‘4’, respectively. The statistical results shown in Table 5 indicated that there were significant differences in the resistance indexes among the three materials. *R. multiflora* ‘13’ was a highly resistant material, *R. multiflora* ‘1’ was a moderately susceptible material, and *R. multiflora* ‘4’ was a highly susceptible material.

In order to further evaluate the resistance of *R**. multiflora* to the powdery mildew, rooted cuttings of *R. multiflora* ‘4’ and ‘13’ were inoculated with powdery mildew. After 10 d, according to grading standards, there were significant differences in the resistance index between *R. multiflora* ‘4’ and ‘13’. *R**. multiflora* ‘13’ was highly resistant and *R**. multiflora* ‘4’ was highly susceptible (Table 6). Based on these results, *R. multiflora* ‘13’ was used as a resistant plant and *R. multiflora* ‘4’ was used as a susceptible plant for subsequent experiments.

### 3.3. Morphological Changes during Rose–Pathogen Interaction

In order to study the developmental changes in the mycelium in the epidermis and the changes of leaf cell structure after the invasion of *R. multiflora* by *P. pannosa,* the differences in leaf tissue structure between the mildew-susceptible and mildew-resistant plants were determined by cytological observation. In susceptible plants, palisade and spongy tissues were underdeveloped, and spongy tissues were arranged sparsely in leaflets. In contrast, the leaflets of resistant plants had well-developed palisade and spongy tissues, and the palisade tissue was closely packed (Figure 4a,b). These differences in the organizational structure of the leaf may partially explain the ability of mildew-resistant roses to withstand *P. pannosa* infection.

At 96 h and 144 hpi, there were pronounced morphological differences between the mildew-resistant and mildew-susceptible roses. In susceptible plants, fungal hyphae had infected the leaf tissue through the stomata on the lower epidermis; papillae were also observed on the upper epidermis (Figure 4c). Hyphae and papillae were not observed in the leaf tissues of the resistant plants, although haustoria were observed at 96 hpi (Figure 4d). Callose accumulation was observed in all susceptible and resistant plants at 96 and 144 hpi (Figure 4e,f).

### 3.4. Statistical Analysis of Transcriptome Sequencing and Gene Expression Profile Results

Through the differential expression analysis of transcripts, it was found that the expression abundance of a large number of transcripts changed under the induction of *P**. pannosa* (Table 7). Difference significance analysis showed that 1916 and 1883 transcripts were differentially expressed in resistant and susceptible plants, respectively, after 24 h of *P**. pannosa* induction, of which 1389 and 1340 genes were annotated, respectively. At 96 hpi by *P**. pannosa*, 1873 and 2481 transcripts were differentially expressed in resistant and susceptible plants, respectively, of which 971 and 1269 genes were annotated, respectively. The differential expression of all genes is shown in the FPKM (fragments per kb per million reads) volcanic distribution map (Figure 5).

This study identified 1504 upregulated transcripts and 396 downregulated transcripts at 24 hpi in resistant plants. In contrast, 258 upregulated transcripts and 1331 downregulated transcripts were identified at 24 hpi in susceptible plants. That is, 78.50% of the differentially expressed transcripts in the resistant plants were upregulated, while only 15.03% of the differentially expressed transcripts in the susceptible plants were upregulated.

At 96 hpi, 818 upregulated transcripts and 1021 downregulated transcripts were identified in the resistant plants. In the susceptible plants, 1549 transcripts were upregulated and 820 transcripts were downregulated at 96 hpi. That is, 43.67% of the differentially expressed transcripts in the resistant plants were upregulated, while 62.43% of the differentially expressed transcripts in the susceptible plants were upregulated (Figure 6).

### 3.5. GO Enrichment Analysis of Differential Transcripts

GO enrichment analysis showed that 327 biological process (BP) functions, 54 cellular component (CC) functions, and 112 molecular function (MF) functions were significantly enriched in the differentially expressed genes in resistant plants invaded by *P**. pannosa* for 24 h. As shown in Figure 7, the differential genes that were significantly enriched in BP functions were related to cell wall organization and biogenesis; those in CC functions were related to the cell wall; and those in MF functions were related to hydrolase activity, chitin binding, and β-galactosidase activity. Compared to resistant plants, 392 BP functions, 73 CC functions, and 188 MF functions were significantly enriched in the differentially expressed genes in the susceptible plants invaded by *P**. pannosa* for 24 h. As shown in Figure 8, the differential genes that were significantly enriched in BP functions were related to the secondary metabolic process, the most significantly enriched CC function was cell wall, and the most significantly enriched MF function was chitin binding.

When the resistant plants were invaded by *P**. pannosa* for 96 h, 513 BP functions, 66 CC functions, and 161 MF functions were significantly enriched in the differentially expressed genes. As shown in Figure 9, the most significantly enriched differentially expressed genes in BP functions were related to secondary metabolic process, hypersensitivity, cell wall organization, and biogenesis. The most significant enrichment of CC functions was detected in the extracellular region, and the most significant enrichment of MF functions was detected in glucosyltransferase activity and transferase activity. Compared to resistant plants, 739 BP functions, 79 CC functions, and 199 MF functions were significantly enriched in the differentially expressed genes in the susceptible plants induced by *P**. pannosa* for 96 h. As shown in Figure 10, the most significantly enriched BP function was related to caprolactam catabolic process, the most significantly enriched CC function was glutamate–cysteine ligase, and the most significantly enriched MF functions were oxidoreductase activity and aldo–keto reductase (NADP) activity.

Comparing the BP functions of differentially expressed genes in susceptible and resistant plants revealed that the differentially expressed genes of susceptible and resistant plants had 79 identical BP functions when invaded by *P**. pannosa* for 24 h, and 254 identical BP functions of differentially expressed genes were found in susceptible and resistant plants when invaded by powdery mildew for 96 h (Figure 11a). GO enrichment analysis showed that when plants were invaded by *P**. pannosa* for 24 h, BP functions that only appeared in resistant plants included the glucan metabolic process, guard cell development, cell metabolic process, and cell disposition in the cell wall. When plants were invaded by *P**. pannosa* for 96 h, BP functions that only appeared in resistant plants included photosynthesis, light harvesting in photosystem II, abscisic acid biological process, and secondary cell-wall biogenesis. When plants were invaded by *P**. pannosa* for 96 h, BP functions including callose deposition in cell wall and cell-wall thickening appeared in both the susceptible plants and resistant plants; however, when plants were invaded by *P**. pannosa* for 24 h, BP functions including callose deposition in cell wall and cell-wall thickening only appeared in the resistant plants and not in the susceptible plants, which showed that the resistance response in resistant plants was faster than that in susceptible plants. These differences may be the reasons why the resistant plants were more resistant to powdery mildew than susceptible plants.

Comparing the CC functions of differentially expressed genes in susceptible and resistant plants showed that the differentially expressed genes of susceptible and resistant plants had 27 identical CC functions when invaded by *P**. pannosa* for 24 h, and 26 identical CC functions were found in susceptible and resistant plants when invaded by *P**. pannosa* for 96 h (Figure 11b). GO enrichment analysis showed that when plants were invaded by *P**. pannosa* for 24 h, CC functions that only appeared in resistant plants included β-galactosidase complex, RNA polymerase complex, SWI/SNF-type complex, microtubule, and 1,3-β-D-glucan synthase complex. When plants were invaded by *P**. pannosa* for 96 h, CC functions that only appeared in resistant plants included photosystem II.

Comparing the MF functions of differentially expressed genes in the susceptible and resistant plants showed that the differentially expressed genes of the susceptible and resistant plants had 23 identical MF functions when invaded by *P**. pannosa* for 24 h, and 59 identical MF functions were found in the susceptible and resistant plants when invaded by *P**. pannosa* for 96 h (Figure 11c). GO enrichment analysis showed that when plants were invaded by *P**. pannosa* for 24 h, MF functions that only appeared in resistant plants included hydrolase activity, β-galactosidase activity, and 1,3-β-D-glucan synthase activity. When plants were invaded by *P**. pannosa* for 96 h, MF functions that only appeared in susceptible plants included β-galactosidase complex.

### 3.6. Expression Analysis of Defense-Related Genes during Infection

To identify the changes in transcription profiles between the resistant and susceptible plants after powdery mildew infection, the level of defense-related gene transcription was analyzed using real-time PCR. The defense-related genes investigated were selected because these genes were identified as differentially expressed in *R. multiflora* during *P. pannosa* infection. Using real-time PCR and a time course experiment, the changes in the transcription levels of the defense-related genes were quantified.

The expression levels of these defense-related genes were significantly different between resistant and susceptible plants, both before and after inoculation (Figure 12). In resistant plants, *kinase* was significantly downregulated 12–144 hpi, while *ERF10* was downregulated 12–96 hpi. In susceptible plants, *kinase* and *ERF10* were both upregulated at 96 hpi. In resistant plants, *chitinase* was upregulated at 12, 24, and 48 hpi, with the highest expression level observed early in the infection process (at 12 hpi). In contrast, *chitinase* was downregulated 1.5-fold in the susceptible plants at 12 hpi; the *chitinase* expression level then steadily increased from 24 to 96 hpi, decreasing again at 144 hpi. In resistant plants, *chitinase* was significantly upregulated 0–48 hpi (2.86-fold); the *chitinase* transcript levels in resistant plants rose more quickly, and to higher maxima, than levels in susceptible plants (Figure 12). These results suggested that *chitinase* was expressed more quickly in resistant plants than in susceptible plants following infection.

At 24 hpi, *callose synthase* was upregulated in both susceptible and resistant plants, indicating that powdery mildew infection caused callose accumulation as a defense response in both susceptible and resistant roses. However, obvious differences in *callose synthase* expression levels were observed between the susceptible and the resistant plants. In resistant plants, *callose synthase* was upregulated and increased steadily from 24 to 144 hpi. However, in susceptible plants, *callose synthase* levels peaked at 24 hpi, decreased at 48 hpi, increased again at 96 hpi, and then decreased at 144 hpi (Figure 12). These results were consistent with the morphological observations.

### 3.7. Expression Analysis of Phenylpropanoid Pathway Genes

The expression levels of genes encoding key enzymes in the phenylpropanoid pathway (*4CL*, *CAD*, *POD*, and *CHS*) were affected by powdery mildew inoculation (Figure 13). In resistant plants, *4-CL* was upregulated at all time points; *4-CL* was also upregulated at all time points, except for at 48 hpi in susceptible plants. In contrast, *CHS* was upregulated in both resistant and susceptible plants between 48 and 144 hpi. The transcription profiles of *CAD* and *POD* differed markedly from those of the other phenylpropanoid biosynthetic pathway genes; the transcription levels of these enzymes were elevated in both resistant and susceptible plants at all time points. These results indicated that gene-encoding enzymes important for the phenylpropanoid pathway were upregulated, and that secondary metabolites played an important role in the interaction between wild roses and powdery mildew.

### 3.8. Expression Analysis of Defense-Related Genes in Response to SA, JA, and ET

Important genes in the SA, JA, and ET pathways were represented in the transcriptome. These genes responded differently in the susceptible and resistant plants. To identify the molecular changes in plants inoculated with *P. pannosa*, the expression levels of *SAG101*, *EDS1*, *PR1*, *PR5*, *NDR1*, *CAT*, *chorismate mutase*, and *PAL* were analyzed.

In the resistant plants, *SAG101* was upregulated from 0 to 48 hpi, with the expression level peaking at 48 hpi (Figure 14). In contrast, *SAG101* expression was obviously inhibited in the susceptible plants from 0 to 48 hpi and was upregulated at 96 hpi. In resistant plants, *EDS1* was rapidly upregulated, reaching peak expression at 12 hpi. In susceptible plants, *EDS1* was upregulated at 96 hpi. *PR1* and *PR5* were upregulated in both resistant and susceptible plants, although both were more highly expressed in the resistant plants. *NDR1* was also upregulated both in resistant and susceptible plants, but the level of expression was similar in both plants. In resistant plants, following inoculation with *P. pannosa*, defense-related genes were induced more quickly than in the susceptible plants. In the resistant plants, *PAL* transcription increased rapidly and peaked at 12 hpi. In susceptible plants, the upregulated levels of *PAL* were lower than those of resistant plants, except for 24–144 hpi. After pathogen inoculation, *CAT* expression was 2–12-fold greater in the resistant plants compared to the susceptible plants. *CAT* was upregulated soon after inoculation in the resistant plants and much later in susceptible plants. The gene expression of *chorismate mutase* was similar in both the resistant and the susceptible plants, and was upregulated from 12 to 96 hpi. In the resistant plants, the *PAL* expression levels were significantly higher than those of the susceptible plants at 12 and 48 hpi.

The expression of *PLD*, *LOX*, and *AOC* genes was affected by *P. pannosa* infection (Figure 14). In the susceptible plants, *PLD* was obviously upregulated at 12 and 24 hpi. In contrast, *PLD* transcript accumulation was only observed in the resistant plants at 48 hpi. *AOC* followed a similar expression pattern, with early upregulation in the susceptible plants (12 hpi) and slower upregulation in the resistant plants (96 hpi). *LOX* was upregulated in both the resistant and the susceptible plants, although the expression levels of *LOX* were greater in the resistant plants than in the susceptible plants at all time points.

In the ET pathway, *Ade* was rapidly upregulated in the susceptible plants (at 24 hpi), and more slowly in the resistant plants (at 96 hpi) (Figure 14). *ACS* was clearly upregulated in the susceptible plants only at 96 hpi, and *ACO* was only clearly upregulated at 144 hpi. This indicated that these genes were expressed in susceptible plants soon after pathogen inoculation. In general, genes associated with the SA pathway were strongly expressed in the resistant plants during *P. pannosa* infection, while the JA and ET signaling pathways predominated in the susceptible plants in order to enhance disease resistance.

## 4. Discussion

This study characterized the morphological and molecular responses of two wild rose varieties (powdery mildew-susceptible and powdery mildew-resistant) to *P. pannosa* infection. Second-generation high-throughput sequencing was used to fully screen the plant transcriptomes during the *R. multiflora*–powdery mildew interaction in order to identify the genes associated with powdery mildew resistance. The temporal and spatial expression levels of some genes important for pathogen defense were analyzed with real-time PCR. Obvious morphological differences were identified between the susceptible and resistant plants, as well as time-dependent differences in resistance-related gene expression patterns.

In plants, the cell wall is a structural barrier of passive defense that also acts as an active and dynamic defense system. The active defense responses of plant cell walls include callose deposition, phenol accumulation, and lignin synthesis [9]. Therefore, cell wall-mediated resistance is an important factor in plant defense systems [27,28,29]. In this study, callose deposition was generally observed in the cell below the pathogen infection site, but occasionally, this deposition encompassed the entire infected cell, physically reinforcing the invasion sites in the cell wall. It has been reported that, in pepper bacterial spot disease, callose deposition is inhibited by the bacterial type III secretion protein, increasing the susceptibility of plants to the disease [30,31,32]. In the present study, callose deposition was observed in the leaves of *R. multiflora* at 96 hpi in the susceptible plants and at 144 hpi in the resistant plants. More callose was deposited in the resistant plants compared to the susceptible plants. In addition, in this study, the differentially expressed genes that were significantly enriched in BP functions were related to the secondary metabolic process, the most significantly enriched CC function was cell wall, and the most significantly enriched MF function was chitin binding in the resistant plants compared to the susceptible plants. This was consistent with research results on the defense response of *Camellia oleifera* against the anthracnose fungal disease [33]. Altogether, these results suggest that callose synthesis and deposition play an active role in plant resistance to powdery mildew.

The phenylpropanoid metabolic pathway has a close relationship with plant disease resistance [34]. The upregulation of gene-encoding key enzymes in the phenylpropanoid pathway increases the resistance of rose plants to powdery mildew [35]. PAL is an important rate-limiting enzyme in the phenylpropanoid pathway that is recognized as the initiator/precursor of benzenoid biosynthesis and deaminates phenylalanine to cinnamic acid [36]. Many studies have shown that as PAL activity increases, plant disease resistance improves; thus, PAL activity can be used as a biochemical indicator of plant disease resistance [37,38]. In this study, *PAL* expression peaked at 12 hpi in resistant plants, and *PAL* was expressed earlier and more strongly in resistant plants compared to susceptible plants.

After pathogenic bacterial infection, the repair of the cell wall is performed through lignification of the plant cell walls, so as to improve the mechanical strength of the cell walls and the plant’s disease resistance [39]. Bhuiyan et al. [40] showed that the lignification of the cell wall played an important role in the resistance of wheat to powdery mildew. In addition, powdery mildew infection upregulated key genes in the lignin synthesis pathway, and silencing these genes significantly reduced the degree of cell-wall lignification in the leaf tissue, increasing the penetration of powdery mildew. In alfalfa, when the shikimate hydroxycinnamyl transferase in the lignin biosynthesis pathway was inhibited, the concentration of resistance-related hormones increased, stimulating the upregulation of related defense genes and increasing the fungal resistance of the alfalfa [41]. CAD, which regulates the direction of phenylpropane metabolism and plays a crucial role in plant growth and development, is another key enzyme in the lignin biosynthesis pathway [42]. Tronchet et al. [43] demonstrated that *CAD* upregulation increased the resistance of *Arabidopsis thaliana* to bacterial spot disease. In the present study, the key genes in the phenylpropanoid synthesis pathway were upregulated in both susceptible and resistant plants, which indicated that different phenylpropanoid pathways were activated after induction. As POD is required for the polymerization reaction at the final step of lignin synthesis, this enzyme is closely correlated with plant disease resistance [44]. Here, the majority of the genes in the lignin biosynthetic pathway (including *chorismate mutase*, *PAL*, *4-CL*, *CAD*, and *POD*) were significantly upregulated after the inoculation of leaves with *P. pannosa*. The expression levels of these genes were higher in the resistant plants than in the susceptible plants. Furthermore, these genes were more quickly and strongly upregulated in resistant plants compared to susceptible plants. This suggests that plant cell wall-mediated disease resistance plays a crucial role in the defense against powdery mildew in roses, and that the upregulation of key genes in the lignin synthesis pathway is one of the most important factors conferring disease resistance to disease-resistant roses. In addition, secondary metabolites in the phenylpropanoid pathway play an important role in the resistance of *R. multiflora* to powdery mildew.

A previous study showed that early morphological resistance was accompanied by a pronounced transcriptional response, including the upregulation of the defense-related genes associated with the SA-, JA-, and ET-mediated defense signal transduction pathways [45], which are the major defense signaling pathways in plants [21]. SA is one of the key signaling molecules that induce SAR in plants and plays an important role in the defense responses that confer resistance [46]. In plants, SAR is indicated by the expression of some disease-related genes (such as *PR1* and *PR5*) [47]. The accumulation of transcripts of these genes generally commences within minutes to hours around the infection sites and within several hours or days at distant sites throughout the whole plant [48]. Endogenous SA induces SAR through transcriptional reprogramming and stimulating immune responses to a broad spectrum of pathogens [49]. SA plays an important role as a signaling molecule during plant defense against pathogens, and increased levels of endogenous SA levels in plants are associated with the induction of many defense-related genes [50,51], including *PR4* and *PR5* [52,53]. In particular, *PR1* is upregulated in response to biotrophic pathogens [54]. After inoculation with *P. pannosa*, the expression levels of *NPR1, EDS1, SAG101, PR1, PR5, CAT*, *chorismate mutase*, and *PAL* increased rapidly, resulting in the rapid synthesis of SA and increasing the acquired resistance of the plant at the early stages of pathogen infection. This was consistent with a previous study on downy mildew in pearl millet [55]. Genes associated with the SA signal pathway were induced in both the resistant and the susceptible plants by powdery mildew. However, these genes were upregulated significantly earlier (*p* < 0.05) and significantly more strongly (*p* < 0.05) in resistant plants.

PLD, LOX, and AOC are key enzymes in the JA synthesis pathway [27,56]. Adenosythomocysteinase, ACS, and ACO are important rate-limiting enzymes in the ET synthesis pathway and are involved in several different steps of the ET synthesis process. Previous studies have shown that plants use different defense signaling pathways to most effectively respond to pathogens [51,57,58,59]. Here, genes associated with the JA and ET signaling pathways were upregulated by powdery mildew infection in both resistant and susceptible plants. In susceptible plants, these genes were expressed earlier and more strongly than in resistant plants. Therefore, the induced genes associated with the JA and ET defense signaling pathways may prevent or delay pathogen attacks in susceptible plants. The SA, JA, and ET defense signaling pathways were all activated during the *R. multiflora*–powdery mildew interaction. The SA pathway was activated earlier and more effectively in the resistant plants, while many genes related to the JA and ET defense pathways were upregulated in susceptible plants (Figure 15).

In summary, this study compared *P. pannosa* fungal penetration, global gene expression, defense-associated genes, and signaling pathway genes between resistant and susceptible varieties of *R. multiflora*. Similar resistance pathways were activated by both resistant and susceptible *R. multiflora* varieties. However, time-course experiments indicated that individual genes were expressed earlier in resistant plants and later in susceptible plants. Furthermore, callose played an important role in the defense reactions of the resistant plants. Both the timing of callose accumulation and the total amount of callose deposited appeared to be critical for optimal disease resistance. The findings of this study increase the understanding of the complex mechanisms underlying the response of *R. multiflora* to powdery mildew and provide a framework for future in-depth characterizations of the pathways and key constituents imparting pathogen resistance.

## Figures and Tables

**Figure 1 genes-13-01003-f001:**
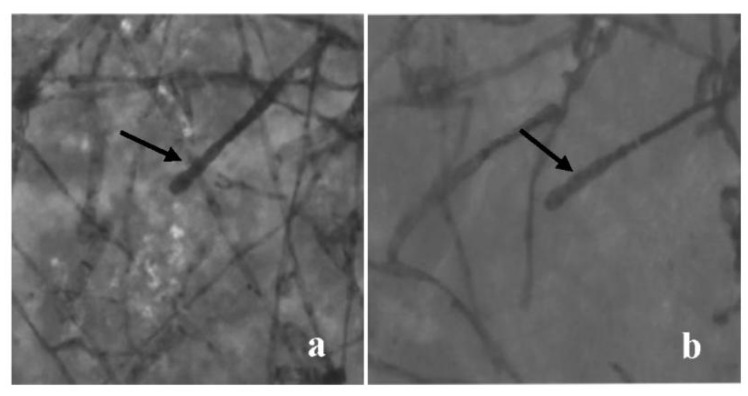
The morphology of powdery mildew fungi. (**a**) The morphology of *Rosa chinensis* powdery mildew fungi, conidiophore (arrow in black). (**b**) The morphology of *Rosa multiflora* powdery mildew fungi, conidiophore (arrow in black).

**Figure 2 genes-13-01003-f002:**
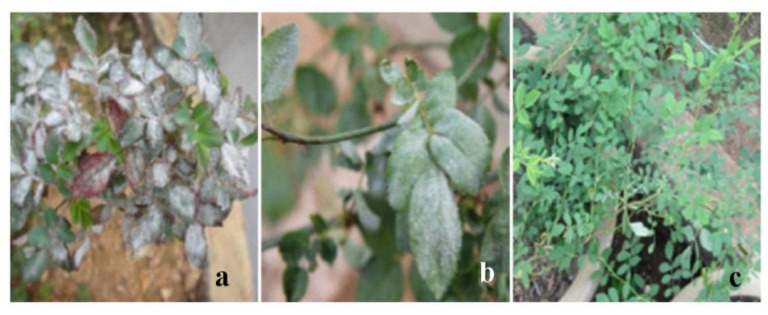
Natural powdery mildew infection on *Rosa multiflora* in the field. (**a**) *R**. multiflora* ‘1’. (**b**) *R**. multiflora* ‘4’. (**c**) *R**. multiflora* ‘13’.

**Figure 3 genes-13-01003-f003:**
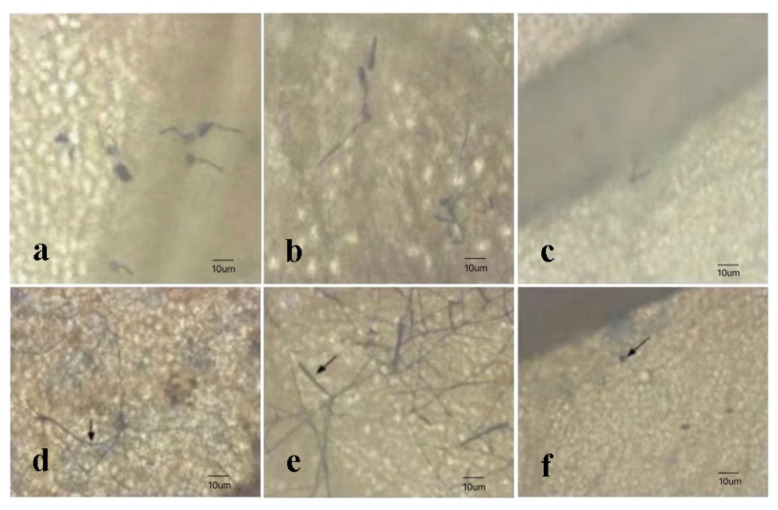
The development process of pathogenic bacteria on the leaves. (**a**) The growth of powdery mildew on *R. multiflora* ‘1’ leaves after being inoculated for 1d. (**b**) The growth of powdery mildew on *R. multiflora* ‘4’ leaves after being inoculated for 1 d. (**c**) The growth of powdery mildew on *R. multiflora* ‘13’ leaves after being inoculated for 1 day. (**d**) The growth of powdery mildew on *R. multiflora* ‘1’ leaves after being inoculated for 4 days, hyphae (arrow in black). (**e**) The growth of powdery mildew on *R. multiflora* ‘4’ leaves after being inoculated for 4 days, conidiophore (arrow in black). (**f**) The growth of powdery mildew on *R. multiflora* ‘13’ leaves after being inoculated for 4 days, ungerminated spores (arrow in black).

**Figure 4 genes-13-01003-f004:**
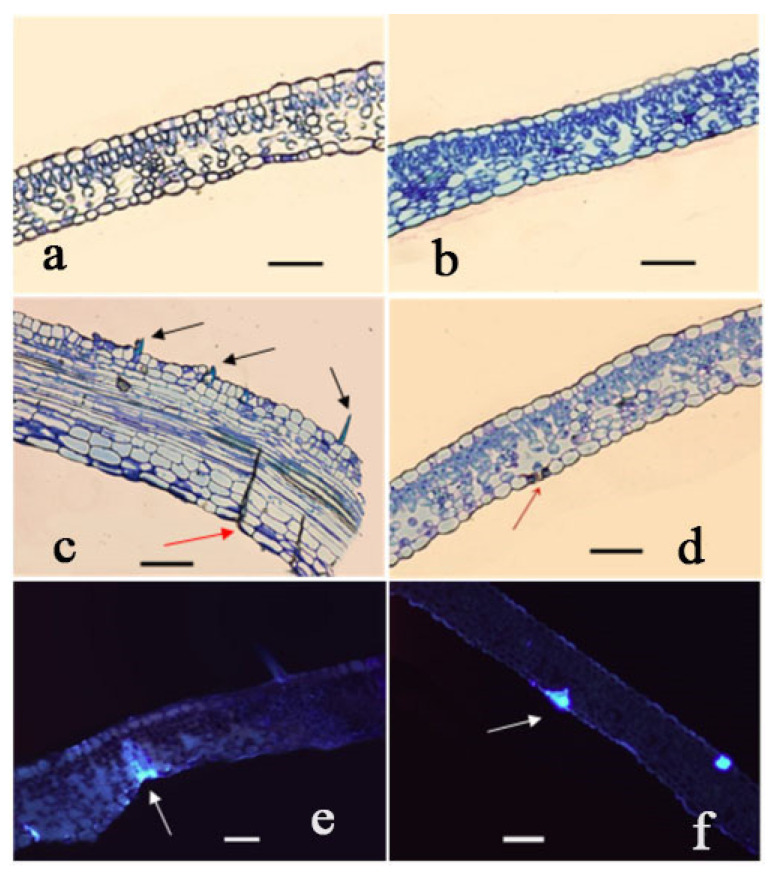
Microscopic observations of interactions between the host and powdery mildew pathogen. (**a**) Uninoculated susceptible plant; bar = 0.2 mm. (**b**) Uninoculated resistant plant; bar = 0.2 mm. (**c**) Invading hyphae (arrow in red) and papillae (arrow in black) in susceptible plant 144 h post inoculation (hpi); bar = 0.2 mm. (**d**) Formation of haustoria in resistant plant 96 hpi (arrow); bar = 0.2 mm. (**e**) Accumulation of callose in susceptible plant 96 hpi (arrow); bar = 0.2 mm. (**f**) Accumulation of callose in resistant plant 96 hpi (arrow); bar = 0.2 mm.

**Figure 5 genes-13-01003-f005:**
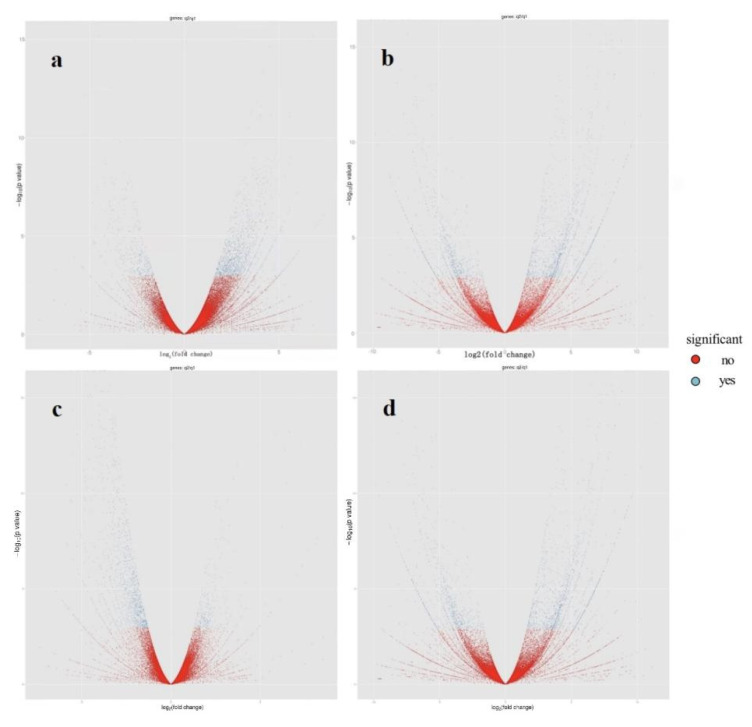
Volcano plot of FPKM of gene expression. (**a**) Resistant plants 24 h after *P**. pannosa* invasion. (**b**) Resistant plants 96 h after *P**. pannosa* invasion. (**c**) Susceptible plants 24 h after *P**. pannosa* invasion. (**d**) Susceptible plants 96 h after *P**. pannosa* invasion.

**Figure 6 genes-13-01003-f006:**
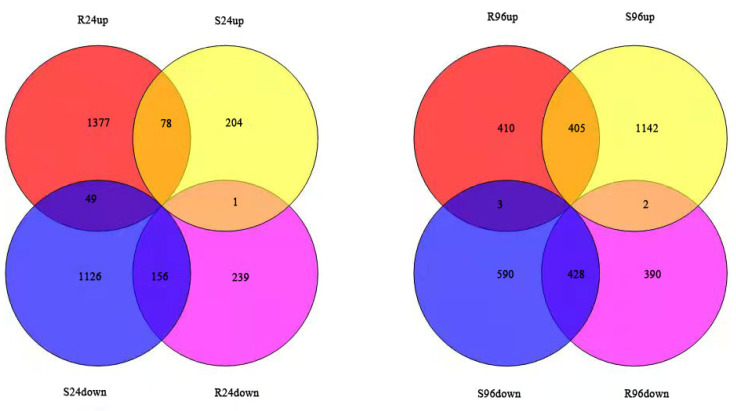
Differentially expressed genes in resistant and susceptible plants after *P**. pannosa* invasion.

**Figure 7 genes-13-01003-f007:**
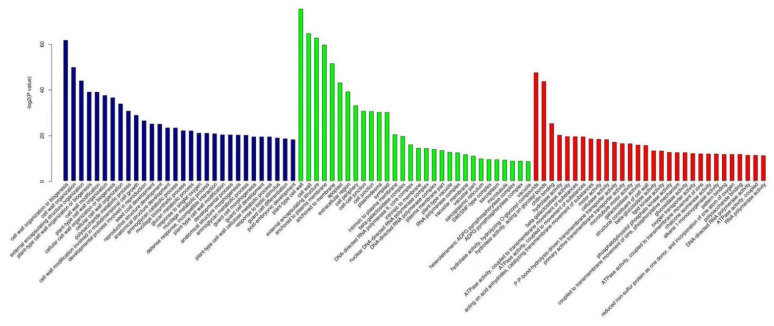
The top 30 functions of differentially expressed genes in resistant plants after being invaded by *P. pannosa* for 24 h.

**Figure 8 genes-13-01003-f008:**
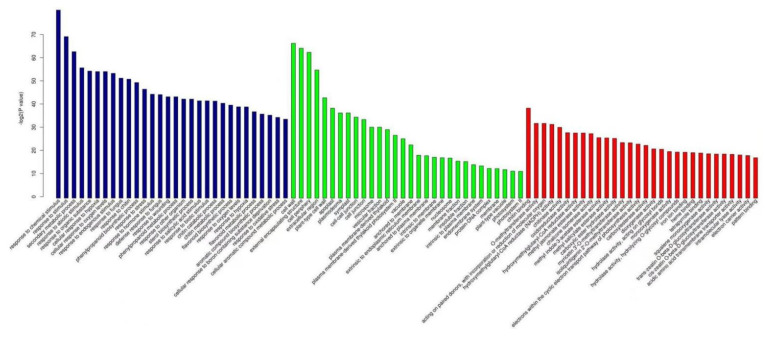
The top 30 functions of differentially expressed genes in susceptible plants after being invaded by *P. pannosa* for 24 h.

**Figure 9 genes-13-01003-f009:**
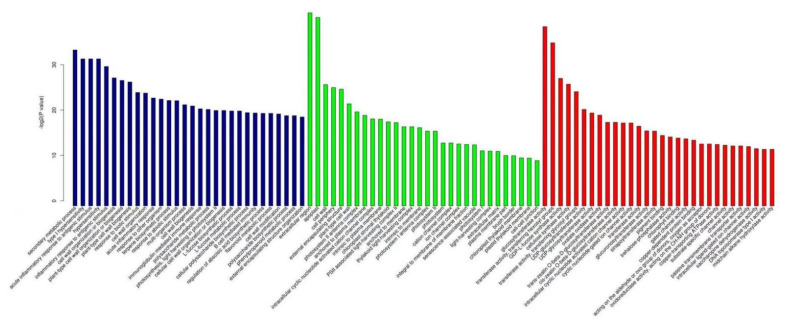
The top 30 functions of differentially expressed genes in resistant plants after beinginvaded by *P. pannosa* for 96 h.

**Figure 10 genes-13-01003-f010:**
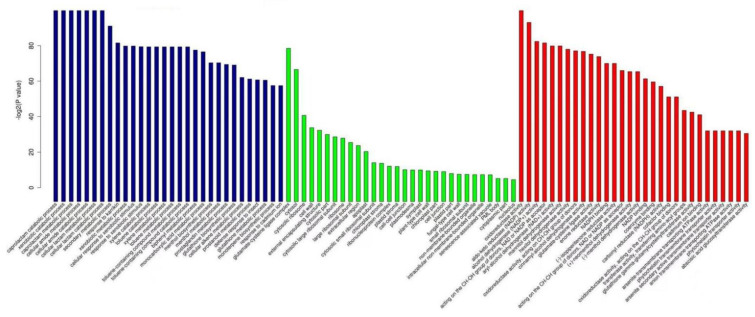
The top 30 functions of differentially expressed genes in susceptible plants after being invaded by *P. pannosa* for 96 h.

**Figure 11 genes-13-01003-f011:**
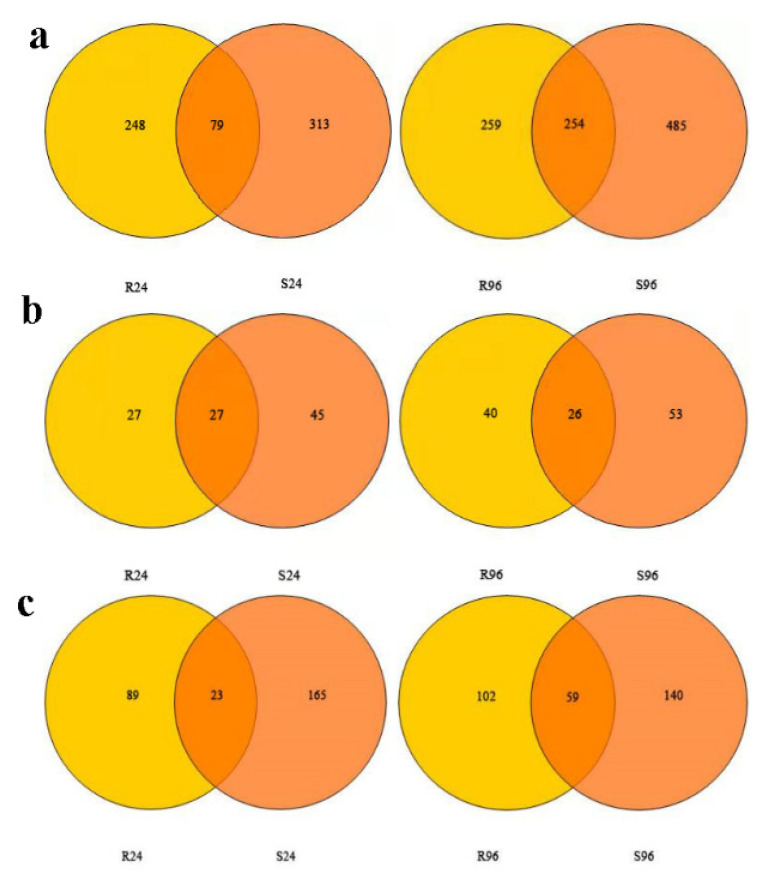
Number of functions of differentially expressed genes in resistant and susceptible plants after invasion by *Podosphaera pannosa*. (**a**) The number of biological process (BP) functions of differentially expressed genes in resistant and susceptible plants after invasion by *P. pannosa*. (**b**) The number of cellular component (CC) functions of differentially expressed genes in resistant and susceptible plants after invasion by *P. pannosa*. (**c**) The number of molecular function (MF) functions of differentially expressed genes in resistant and susceptible plants after invasion by *P. pannosa*.

**Figure 12 genes-13-01003-f012:**
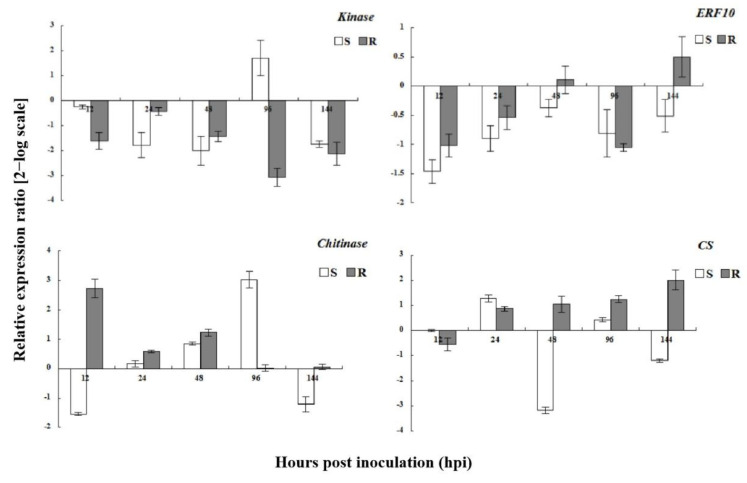
Expression analysis of defense-related genes in susceptible and resistant plants after *P**. pannosa* inoculation.

**Figure 13 genes-13-01003-f013:**
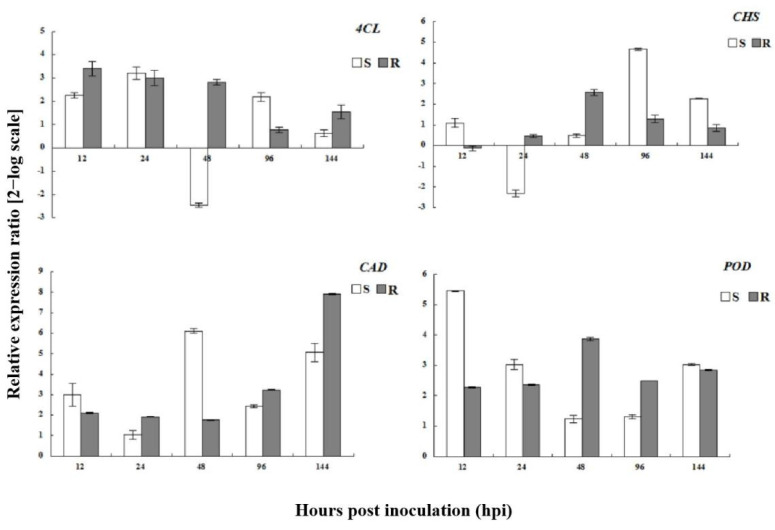
Expression analysis of phenylpropanoid pathway genes in susceptible and resistant plants after *Podosphaera pannosa* inoculation.

**Figure 14 genes-13-01003-f014:**
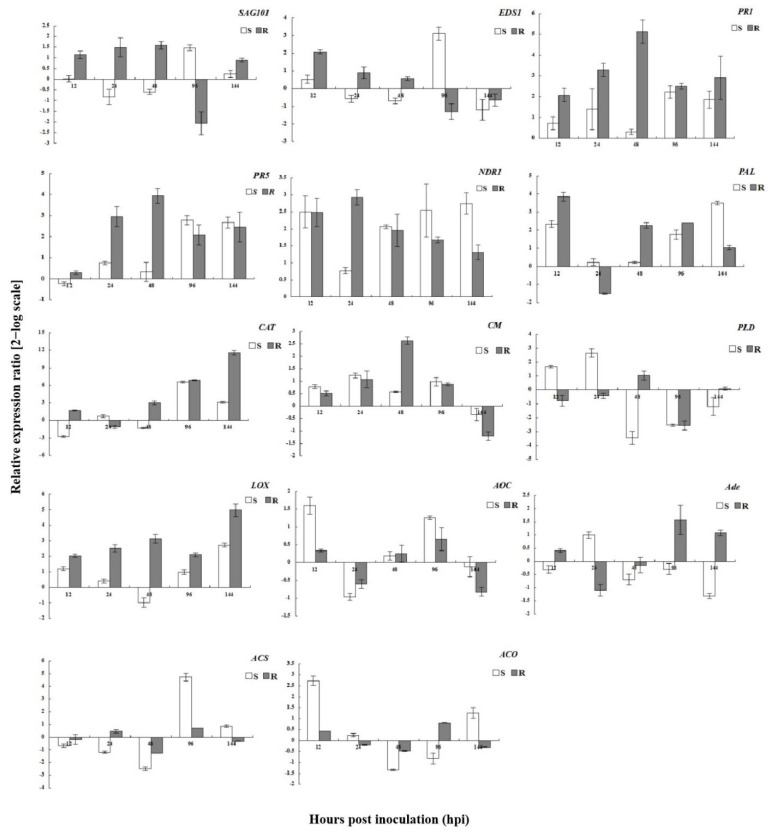
Expression analysis of defense-related genes in response to salicylic acid (SA), jasmonic acid (JA), and ethylene (ET) in susceptible and resistant plants after *Podosphaera pannosa* inoculation.

**Figure 15 genes-13-01003-f015:**
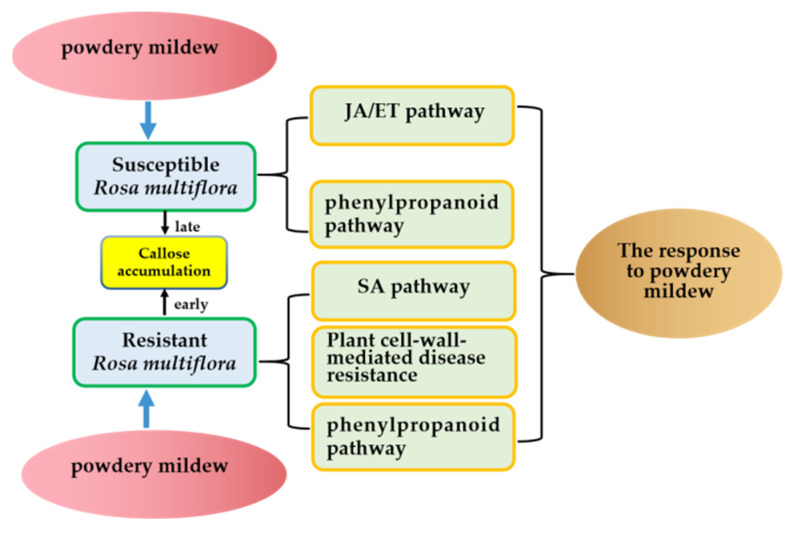
Schematic model for resistance mechanism of *R. multiflora* to powdery mildew.

**Table 1 genes-13-01003-t001:** Grading standard of powdery mildew.

Disease Grade	Incidence Degree
Grade 0	There was no powdery mildew on the leaves of the whole plant
Grade 1	One to two leaves had thin hyphae
Grade 2	Three to four leaves had medium mycelium and some spores
Grade 3	Five to six leaves had thick hyphae and more spores
Grade 4	More than seven leaves had large spore piles and a large number of spores

**Table 2 genes-13-01003-t002:** Standard of micro-observation methods.

Disease Grade	Mycelial Growth Rate and Disease Grade Classification Standard	Classification Criteria of Mycelial Coverage Area and Conidial Stalk Formation
Grade 0	Spores did not germinate	No hyphae, some spores did not germinate
Grade 1	Average mycelium length of 0–7 μm	A small amount of mycelium; the hyphae formed by the germination of one spore covered 0–1/20 of the leaves
Grade 2	Average mycelium length of 7–14 μm	Numerous hyphae; the hyphae formed by the germination of one spore covered 1/20–1/10
Grade 3	Average mycelium length of 14–21 μm	Numerous hyphae; the hyphae formed by one spore germination covered more than 1/10 of the leaves, but no conidiophores were observed
Grade 4	Average mycelium length of more than 21 μm	Numerous hyphae; the mycelium grew fast, and conidiophores were formed

**Table 3 genes-13-01003-t003:** Standard of powdery mildew on rooted cuttings.

Disease Grade	Standard
Grade 0	No pathogens were observed
Grade 1	Leaf area occupied by pathogens <l%
Grade 2	Leaf area occupied by pathogens l–5%
Grade 3	Leaf area occupied by pathogens 6–20%
Grade 4	Leaf area occupied by pathogens 21–40%
Grade 5	Leaf area occupied by pathogens 41–60%
Grade 6	Leaf area occupied by pathogens >60%

**Table 4 genes-13-01003-t004:** *Rosa multiflora* powdery mildew resistance levels in the field.

Materials	Resistance Index	Resistance Level
*R. multiflora* ‘13’	0.85000	high resistance
*R. multiflora* ‘1’	0.06250	high susceptibility
*R. multiflora* ‘4’	0.08750	high susceptibility

**Table 5 genes-13-01003-t005:** *Rosa multiflora* powdery mildew resistance levels obtained with the micro-observation method.

Materials	Resistance Index	Significance of Difference	Resistance Level
*R. multiflora* ‘13’	0.84167	A	high resistance
*R. multiflora* ‘1’	0.39167	B	moderate susceptibility
*R. multiflora* ‘4’	0.04167	C	high susceptibility

**Table 6 genes-13-01003-t006:** Powdery mildew resistance levels of *Rosa multiflora* grown from cuttings.

Materials	Resistance Index	Resistance Level
*R. multiflora* ‘13’	0.90500	high resistance
*R. multiflora* ‘4’	0.02400	high susceptibility

**Table 7 genes-13-01003-t007:** Number of differentially expressed genes after *Podosphaera pannosa* invasion.

Treatment Number of Transcripts	Differential Expression Multiple log2 Value > 5 or <−5	Differential Expression Multiple log2 Value > 7 or <-7	Differential Expression Multiple log2 Value > 9 or <−9
Resistant plants 24 h post-inoculation (hpi)	62	7	0
Resistant plants 96 hpi	531	132	16
Susceptible plants 24 hpi	63	8	0
Susceptible plants 96 hpi	916	189	35

## Data Availability

Not applicable.

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
