# Peer review of "Morphological and Molecular Analyses of the Interaction between *Rosa multiflora* and *Podosphaera pannosa"

_genes, 2022, doi:10.3390/genes13061003_

Round 1
Reviewer 1 Report
The authors have investigated the defense responses of two rose genotypes (highly resistant vs. highly susceptible). Unravelling effective molecular mechanisms against powdery mildew is an interesting subject for a wide range of readers, as the disease is one of the most economically important diseases of the industry. Although the presented results are informative and valuable, the manuscript requires some major edits to merit publication. I have listed some examples of the possible edits here, but the list is longer than this as there are several other areas with similar situations:
1- The abstract seems to be written in a hasty manner. The writing is not well composed, and some information is missing. For example: line 13 and 14, why Cytological and Transcriptome are capitalized?; line 15, level should be plural (levels); line 16, should be "96 and 144 h after inoculation"; line 19, should be "the differentially expressed genes"; line 22, why capitalized chitinase?; line23, prior introduction of the species is missing; line24, pathway should be plural; line 27, present tense is needed for "demonstrate" here as a conclusion is being made.
2- Material and Methods: line 93, which "was" in good growth? a verb seems to be missing; line 97, "plants using" (space needed) ; line 97: past tense is needed there "Each treatment included..."; line 105: how do you know that the lack of susceptibility is resistance? it might be due to lack of successful infection or any other problem during the host and pathogen interaction. So, converting Disease Index to Resistance Index simply by doin 1-DI (although mathematically correct) is not pathologically justifiable; line 124, treatments should be inoculation?; line 127, it is needed to briefly add some information about the staining protocol and the type of dyes used; line 135, similarly, it is important to briefly mention some information about the bioinformatic pipeline used, what package was used for read alignment? what package for transcript assembly and differential gene expression analysis? where is the code in case any reproducibility attempt would be needed; Table 3 could be presented as a supplementary information.
2- There are several places where an introductory sentence has been added to the beginning of the result section/paragraph. All those should be transferred to the Introduction. There are some "conclusions" or "interpretation" of the results in there as well (like line 370). Those parts need to be moved to the Discussion section. Results should report only Results and observation.
Some lines and words have different sizes of font like line 223, 264, 265, 267 and several other lines.
Figure 5 is not readable. I recommend providing a table for this part. This is the most important part of the RNAseq analysis and should be presented very clearly (which is not in the current format).
All the introductory/interpretations in the Result section should be moved to their proper section.
3- Discussion: it is generally well written. I recommend that the authors add a schematic model at the end to better show their SA/JA mechanistic story.
A couple of fundamental questions would be:
Why those differentially expressed genes in the RNAseq part were not tested with qPCR? Why defense related genes were separately chosen here? why the defense related genes were not discovered by RNAseq (if they were found to be differentially expressed by qPCR)?
Reviewer 2 Report
The research paper compared the plant responses to powdery mildews between two different wild rose varieties including the morphological and molecular changes. The research idea is good however the manuscript design and writing have major revision points:
- The introduction section needs more details about the powdery mildew disease, its causes, and its significance.
- The authors didn’t write anything about pathogen source, isolation, or even its identification in material and methods.
- Microscopically description or genetic identification of the pathogen must include.
- Authors mentioned that (Conidiophores from the pannosa were employed as inoculum); how did the authors separate the fungal conidiophores from the fungal mycelia and spores.
- Then authors mentioned that inoculum was 104 conidispores/ml (Is it conidiophores or spores?).
- To clarify the pathogenicity stages scanning and transmission electron microscope are recommended.
- The manuscript English writing needs serious revision by a native speaker.
Round 2
Reviewer 1 Report
Yes, I am satisfied with the provided edits and revisions.
Reviewer 2 Report
The authors give satisfying answers and add the missing research parts. So, I recommended it for publishing.